# Guiding of relativistic electron beams in dense matter by laser-driven magnetostatic fields

M. Bailly-Grandvaux[1], J.J. Santos [1], C. Bellei[1], P. Forestier-Colleoni[1], S. Fujioka [2], L. Giuffrida[1], J.J. Honrubia[3], D. Batani[1], R. Bouillaud[1], M. Chevrot[4], J.E. Cross[5], R. Crowston[6], S. Dorard[4], J.-L. Dubois[1], M. Ehret[1,7], G. Gregori[5], S. Hulin[1], S. Kojima[2], E. Loyez[4], J.-R. Marquès[4], A. Morace[2], Ph. Nicolaï[1], M. Roth[7], S. Sakata[2], G. Schaumann[7], F. Serres[4], J. Servel[1], V.T. Tikhonchuk[1], N. Woolsey[6] & Z. Zhang[2]

Intense lasers interacting with dense targets accelerate relativistic electron beams, which transport part of the laser energy into the target depth. However, the overall laser-to-target energy coupling efficiency is impaired by the large divergence of the electron beam, intrinsic to the laser–plasma interaction. Here we demonstrate that an efficient guiding of MeV electrons with about 30 MA current in solid matter is obtained by imposing a laser-driven longitudinal magnetostatic field of 600 T. In the magnetized conditions the transported energy density and the peak background electron temperature at the 60-μm-thick target's rear surface rise by about a factor of five, as unfolded from benchmarked simulations. Such an improvement of energy-density flux through dense matter paves the ground for advances in laser-driven intense sources of energetic particles and radiation, driving matter to extreme temperatures, reaching states relevant for planetary or stellar science as yet inaccessible at the laboratory scale and achieving high-gain laser-driven thermonuclear fusion.

[1] Univ. Bordeaux, CNRS, CEA, CELIA (Centre Lasers Intenses et Applications), UMR 5107, F-33405 Talence France. [2] Institute of Laser Engineering, Osaka University, 2-6 Yamada-oka, Suita, Osaka 565-0871, Japan. [3] ETSI Aeronáutica y del Espacio, Universidad Politécnica de Madrid, Plaza del Cardenal Cisneros 3, Madrid 28040, Spain. [4] LULI, UMR 7605, CNRS, Ecole Polytechnique, CEA, Université Paris-Saclay, UPMC: Sorbonne Universités, F-91128 Palaiseau cedex France. [5] Department of Physics, University of Oxford, Parks Road, Oxford OX1 3PU UK. [6] Department of Physics, University of York, Heslington YO10 5DD, UK. [7] Institut für Kernphysik, Technische Universität Darmstadt, Schlossgartenstrasse 9, 64289 Darmstadt Germany. Correspondence and requests for materials should be addressed to J.J.S. (email: joao.santos@u-bordeaux.fr)

Optimization of high-energy-density (HED) electron flows through solid-density or denser matter is a major challenge for improving laser-driven sources of energetic particles and radiation[1–3], or for driving matter to temperatures relevant to the study of structural and dynamic properties of warm dense matter or HED matter[4] of interest in planetary science[5,6] or astrophysics[7,8], or even for the development of high-gain inertial confinement fusion (ICF) schemes[9,10]. When interacting with dense (opaque) targets, intense laser pulses drive high-current relativistic electron beams (REB), which can transport a significant fraction of the laser energy into the targets's depth[11–13]. However, the energy-density flux degrades rapidly against the penetration depth due to resistive and collisional energy losses[14–18] and mostly to the intrinsically large divergence of the REB[19–21], as a result of the laser–plasma interaction and the development of electromagnetic instabilities at the target surface[22,23]. Devising means of controlling the REB transverse spread and confine its propagation within a small radius would for example maximize the electrostatic energy exchange with the ions and their acceleration[24], or optimize electron energy transport and isochoric matter heating[4], effects of great benefit in the aforementioned research fields and applications. For example, in the framework of the Fast Ignition (FI) scheme for ICF[25,26], imposed axial magnetic fields (B-fields) in the 1–10-kT range should be able to guide GA currents of MeV electrons over 100-μm distances from the laser-absorption region, i.e., up to the dense core of nuclear fuel[27,28]. This would enhance the electrons energy coupling to the core, potentially leading to high-gain fusion-energy release while reducing the needed ignitor-laser energy.

Radially confined REB transport has been experimentally reported due to self-generated resistive B-fields by using specific laser irradiation schemes and/or target structures[29–33]. The common principle is that collimating B-fields are self induced by REB intense currents in resistive media, due either to the current shear or to radially converging gradients of resistivity along the REB propagation axis[34,35]. Nonetheless, so far obtained data show that many electrons are not magnetically trapped and maintain their initial divergence and radial spread. The number of guided electrons remains under 25%. Proposed improvements for REB self-guiding involve sophisticated target structures[36–38], which could hardly be tested in the harsh conditions of an ICF target.

In the present work we apply an external B-field in REB transport experiments. The ≈600-T B-field is enough strong to guide MeV-range electrons in solid targets. It is produced by an all-optical technique using laser-driven coil targets[39–46]. This technique creates a magnetostatic field of sufficiently long duration to fully magnetize the transport target prior to REB generation. Our results clearly show efficient REB pinching through solid-density targets of 60-μm thickness. Benchmarked simulations reveal about a factor of five increase in energy-density flux at the target's rear surface.

## Results

**Experimental setup.** The experiments were conducted at the LULI pico 2000 laser facility with a 1.06-μm wavelength ($1\omega_0$) dual laser beam configuration: A high-energy long-pulse beam (LP: 1 ns, $500 \pm 30$ J, $(1.4 \pm 0.6) \times 10^{17}$ W cm$^{-2}$) focused into Ni coil-targets produced a B-field of several hundred Tesla and duration of a few nanoseconds[42]. At different delays $\Delta t$ with respect to the LP-laser, a high-intensity short-pulse beam (SP: 1 ps, 47–49 J, $(1.5–3) \times 10^{19}$ W cm$^{-2}$) was focused at normal incidence and generated a REB in solid targets. The setup at the coil vicinity is sketched in Fig. 1a. Further details on the coil-targets geometry and laser irradiation are given in Methods and Supplementary Note 1.

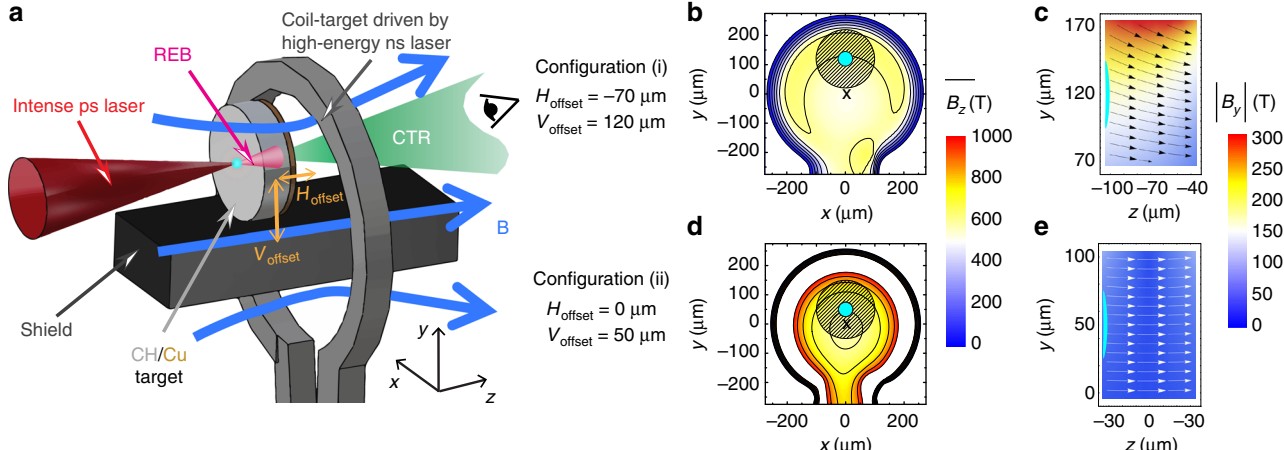

**Fig. 1** Experimental configuration for relativistic electron beam transport with imposed B-field. **a** Sketch of the experimental setup at the coil vicinity: the relativistic electron beam (REB) is generated by the intense ps laser, focused parallel to the coil axis and at normal incidence onto the center of the front surface of a neighboring solid 50-μm-CH/10-μm-Cu-thick target of 200-μm diameter. An intense current discharge is previously driven in the Ni coil target (coil radius of 250 μm) by a high-energy ns laser, yielding a dipole-like B-field along the coil axis. REB patterns were investigated by imaging the coherent transition radiation (CTR) emitted from the transport targets' rear surface. **b–e** B-field distribution in vacuum (origin of the spatial coordinates at the coil center) at its peak value, 1 ns after ns-laser driving, as experimentally and numerically characterized in ref. [42]. **b, d** Amplitude of the B-field longitudinal component averaged over the 60 μm target thickness, $\overline{B_z}$, at the two explored positions of the transport target (offsets of the target center with respect to the coil center, as labeled). The dashed circles represent the position of the transport target in the perpendicular plane. The coil axis and the intense ps-laser axis are respectively represented by the cross signs and the center of the light-blue circles. The radius of the later corresponds to the REB source radius at the ps-laser-irradiated target surface, $r_0$, in the REB-transport simulations. **c, e** Absolute value of the B-field vertical component $|B_y|$ (color scale) and arrow representation of the B-field lines over the target $x = 0$ -slice, for the two target positions. The plots correspond also to the B-field embedded into the targets as initial conditions for the REB-transport simulations in magnetized conditions, in agreement with predictions of the B-field resistive diffusion. The light-blue regions on the left-hand side of the plots indicate the size of the REB source. Image in **a** was created by M.B.-G. and J.J.S.

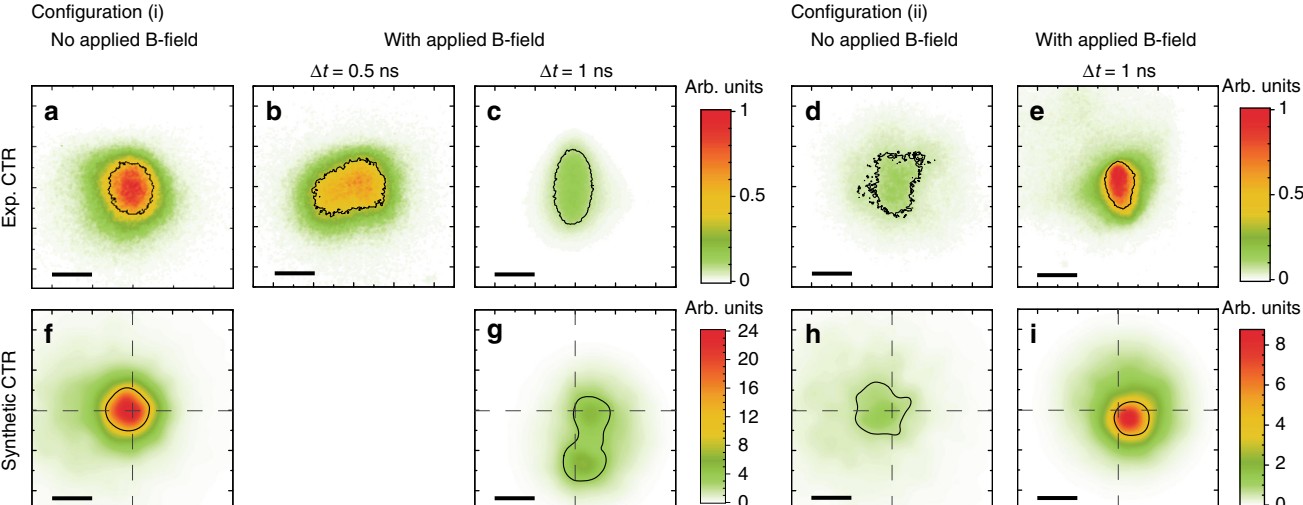

**Fig. 2** Experimental and synthetic images of the coherent transition radiation. Coherent transition radiation (CTR) is produced at the transport targets' rear surface when relativistic electrons cross the boundary between the target and vacuum. **a–e** Experimental data and **f–i** synthetic CTR calculated from 3D PIC-hybrid simulations of fast electron transport, for the two configurations (i) target out of the coil plane and (ii) target at the coil plane (Fig. 1), with and without imposed B-field. The black horizontal bars, corresponding to 20 μm, give the spatial scale at the emitting target surface. The contour lines correspond to the half-height of the signals. The crossed dashed lines indicate the position of REB injection at the targets' front laser-irradiated surface. The synthetic images are calculated at the end of the simulation runs and account for all particles having reached the targets' rear surface

The REB transport targets were 200-μm-diameter and 50-μm-thick plastic (CH) cylinders with a 10-μm-thick Cu coating on the rear side. The cylinder's axis was invariably parallel to the coil axis, and for the two experimental runs, we explored successively positioning the target in two configurations: (i) shifted from the coil plane (with horizontal and vertical offsets of the target center with respect to the coil center of $H_{offset} = -70$ μm, $V_{offset} = 120$ μm), and (ii) at the coil plane ($H_{offset} = 0$ μm, $V_{offset} = 50$ μm). This enabled us to explore two different 3D spatial distributions for the B-field imposed to the transport targets, as seen in Fig. 1b, c for configuration (i) and Fig. 1d, e for configuration (ii). For each of the two configurations, the choice of $\Delta t$ controlled the time allowed for B-field diffusion in the transport targets prior to REB injection, testing REB transport in different conditions of target magnetization.

The evolution of the transport-target magnetization has been predicted by simulations of the B-field resistive diffusion inside the target as the B-field rises up to its peak value (rise-time of ≈1 ns, consistent with the duration of the LP-laser driver). The results show that by ≈1 ns the transport targets are fully magnetized: the B-field spatial distribution inside the target is then similar to the distribution expected in vacuum at the target position (Supplementary Note 2). Assuming a constant resistivity $\eta = 10^{-6}$ Ωm (expected for CH at 1 eV), this magnetization time agrees with a simple linear estimate of the B-field diffusion time $\tau_{diff} = \mu_0 L^2/\eta \approx 1$ ns over the length $L = 50$ μm of the target CH layer.

The REB transverse pattern after crossing the target thickness was investigated by imaging the Coherent Transition Radiation (CTR) emission from the rear surface at twice the laser frequency, $2\omega_0$. The emitting surface was imaged at a 22.5° horizontal angle from the target normal into an optical streak camera used with a wide slit aperture as a fast gated 2D frame grabber[19]. CTR is a non-linear emission mechanism providing the signature of relativistic electrons at their first transit through the target's rear surface[16,47–49].

**Experimental results**. Sample results of the CTR signals are shown in the first row of Fig. 2, for target position configurations

(i) on the left, and configuration (ii) on the right, with and without imposing an external B-field, as labeled. The aspect ratio of the signals has been corrected from the observation angle. For the two data sets, the average SP laser energy and intensity were respectively (i) $47 \pm 6$ J and $3.0 \pm 0.8 \times 10^{19}$ W cm$^{-2}$, (ii) $49 \pm 1$ J and $1.5 \pm 0.4 \times 10^{19}$ W cm$^{-2}$. The difference in laser intensity is mainly due to different focal spots in the two independent experimental runs.

Without externally imposed B-field (Fig. 2a, d), we obtained rather large (≈14 ± 2 μm half-width-half-maximum, HWHM) and fairly symmetric CTR patterns. When imposing the longitudinal B-field for the target position (i) and varying the delay of REB injection, at $\Delta t = 0.5$ ns (Fig. 2b) the CTR yield is slightly weaker and its pattern looks twisted, yet the average size is comparable to the case without B-field. As mentioned before, the target should not be yet fully magnetized.

At $\Delta t = 1$ ns we have obtained CTR patterns significantly different than the case without B-field, for both target positions (i) and (ii) (respectively Fig. 2c, e). The CTR patterns are clearly narrower horizontally. Vertically, the signal is also narrower for configuration (ii), while it is elongated for configuration (i). These correspond to half-height areas of equivalent radius ≈13 μm for configuration (i) and ≈9 μm for configuration (ii). The CTR yield decreased for configuration (i) and increased for configuration (ii) relative to the corresponding signals without B-field. As discussed above, the delay $\Delta t = 1$ ns corresponds to REB transport in magnetized targets. As for a 600 T field (see values of $\overline{B_z}$ around the REB axis in Fig. 1b, d) the Larmor radius of MeV electrons becomes smaller than the REB initial radius, the electrons are in principle trapped and follow the B-field lines. The differences in the patterns shape and yield between Fig. 2c, e are then related to the different B-field spatial distributions inside the targets, as represented in Fig. 1c, e. In particular, the CTR yield drop in configuration (i) is explained further below by means of REB transport simulations: the $B_y$ component (Fig. 1c) deviates downwards the REB propagation axis (further away from the solid angle of the CTR lens), elongates the REB path inside the target and broadens its momentum distribution. In comparison, for configuration (ii) with optimum B-field symmetry, the REB

axis is not deviated and the narrowing of the signal in Fig. 2e compared to Fig. 2d is significant, more symmetric, and is a more clear signature of a radially pinched REB. An increase in the REB density at their first transit through the target's rear surface is here directly evidenced by an enhancement of the CTR yield by a factor 6.

**Relativistic electron transport modeling.** Details of the REB propagation and energy transport were unfolded by simulations using a 3D particle-in-cell (PIC)-hybrid code for the electron transport[50], accounting for fast electron collisions with the background material and REB self-generated fields. We simulated REB transport in both magnetized and unmagnetized conditions. For the former case, we assumed full target magnetization (corresponding to $\Delta t = 1$ ns) imposing as initial conditions the 3D distributions illustrated in Fig. 1b–e for the B-field embedded into the targets. The initial REB total kinetic energy was set to 30% of the on target SP-laser energy, and injected at the front surface over a region of $r_0 \approx 25$-µm-radius HWHM (light-blue marks in Fig. 1b–e), corresponding to empiric factors 4 or 3 of the SP-laser focal spot size HWHM, respectively, for configuration (i) or (ii). The injected electron kinetic energy spectra (dashed lines in Fig. 3a, b) were characterized by power laws for the low energy part $\propto \varepsilon_k^{-1.6}$ and exponential laws for the high energy part $\propto \exp(-\varepsilon_k/T_h)$ with $T_h^{i)} = 2.0$ MeV and $T_h^{ii)} = 1.3$ MeV, as predicted by the ponderomotive potential[51] for the corresponding laser parameters. The injected angular distribution was characterized by a 30° mean divergence angle and a 55° dispersion angle as defined in refs [23,38,52]. All the above geometric and energy REB source parameters are consistent with our previous experimental and numerical characterization for the data obtained in the same laser facility using equivalent laser parameters[17,52] (see Supplementary Note 3 for further details). The total simulation time was set to 3.6 ps (with $t = 1.25$ ps corresponding to the peak REB flux at the front surface).

For a direct comparison with the experimental data, we developed a synthetic CTR-emission post-processor applied to the transport code output. CTR is reconstructed by the coherently added transition radiation fields produced by each simulated macroparticle[49]. Details on the parameters and assumptions of the CTR post-processing are presented in Methods and in Supplementary Note 4. Synthetic CTR signals are presented in Fig. 2f–i, reproducing fairly well the experimental CTR patterns as well as the relative signal yield change when imposing the B-field. In more details, the simulations reproduce with $15 \pm 2\%$ relative errors the ratio of CTR yield (with B-field/without B-field) for both target positions. Regarding the patterns, the synthetic CTR in the magnetized conditions is, in fair agreement with the experimental data, radially pinched for configuration (ii) (Fig. 2i) and vertically elongated for configuration (i) (Fig. 2g). The experimental patterns' radius (azimuthally averaged), with or without B-field, is reproduced with $15 \pm 5\%$ relative errors, except for the magnetized case of configuration (i), where the elliptic shape of the experimental signal is not exactly the same of the synthetic one and the relative error amounts to $\approx 33\%$.

As for the corresponding REB features and energy transport, Fig. 3 shows the simulation results at the 60-µm-thick targets' rear side surface for the two target position configurations: Fig. 3a, b show the time-integrated electron energy spectra for simulations with (red) and without (blue) B-field, compared to the corresponding spectrum at the front surface (dashed black). A significant number of electrons with energy $\varepsilon_k < 100$ keV are absorbed or scattered out of the simulation box before crossing the target, as expected from the direct collisions with the background material and by electric-field stopping linked to the

resistive neutralizing current of thermal electrons[17,18,52]. These losses are slightly mitigated with the symmetric B-field, configuration (ii). The time-integrated REB energy-density flux patterns (Fig. 3c–f) clearly show radial pinching due to the imposed B-fields, increasing the peak values by factors $\approx 15$ and $\approx 20$ and decreasing the beam mean radius by factor $\approx 3$ and $\approx 2$, respectively, for configurations (i) and (ii). An other positive outcome is that the imposed B-field smoothes the REB filaments compared to simulations without B-field, for both target positions. The substantially smoother, narrower and denser beams in Fig. 3d, f correspond to unprecedented efficient guiding and improved energy-density flux. The impact is clearly seen in the reached peak background electron temperature (Fig. 3g–j), which is higher with B-field by a factor $\approx 5.9$ for both target positions.

Yet, it is noticeable in the simulations that the B-field in configuration (i) deviates vertically the REB, exiting the target with a vertical shift of $\approx 23$ µm and an angle of $\approx 20°$ with respect to its injection horizontal axis. This deviation increases and broadens the electrons' transport time and momentum-angle distribution at the target rear side, contributing to roughly half of the CTR-yield drop (Fig. 2c, g) compared to the unmagnetized transport case (Fig. 2a, f). The vertical deviation also directs the REB away from the CTR collecting lens. Calculations on how the CTR yield would depend on the collecting lens angular position are given in Supplementary Note 4 and Supplementary Fig. 3 and lead to conclude that the setup lens position accounts for the other half of the total CTR signal drop.

The efficient magnetic REB guiding is further evidenced by analyzing the phase-space maps of the electrons reaching the targets' rear surface. For configuration (ii), the transverse horizontal coordinates phase space $(x, p_x)$ is plotted in Fig. 4: in Fig. 4a without B-field, the inclined shape of high ellipticity is characteristic of symmetric correlated transverse momentum and position of a regularly diverging beam; in Fig. 4b with B-field, the phase-space map is significantly narrower than in Fig. 4a, but only for the spatial-coordinates. The induced cyclotron effect de-correlates the electrons positions and momenta and their radial spread is limited as they are trapped and move along (rotating around) the B-field lines. This is possible as the $\approx 10$-µm Larmor radius, calculated for the REB-source mean kinetic energy $\overline{\varepsilon}_k \approx 1$ MeV and a 600-T B-field, is smaller than the REB-source radius $r_0 \approx 25$ µm. Yet the B-field does not really affect the electrons intrinsic divergence as the width of the transverse momenta distribution is maintained.

To understand the way the energy is forwardly transported into the target, we plot in Fig. 4c the evolution with target depth of the time-integrated REB total kinetic energy ($W_k$, diamonds connected by dashed lines) and REB energy encircled over the surface corresponding to the initial REB source, $\pi r_0^2$, kept centered with the injection axis ($W_{k-r_0}$, stars connected by solid lines). The loss rate of $W_k$ against target depth is comparable for the two cases without (blue) and with (red) B-field, except for the first ~10 µm where the B-field efficiently confines electrons and also smoothes the REB filaments. About 45% more energy is transported to the target rear in the magnetized case due to the magnetic confinement mitigating the high diffusivity of low-energy particles. Much more importantly, the $r_0$-encircled energy around the injection axis at the target rear in the magnetized case contains $\approx 66\%$ of the total transported energy, against only $\approx 18\%$ for the unmagnetized case. As a consequence, the time-integrated REB energy-density flux after crossing the target thickness increases by $\approx 5.3\times$ by applying the B-field, as visible from the comparison of Fig. 3e, f.

In Fig. 5, two key experimental CTR metrics (horizontal dashed bars) are compared to the corresponding synthetic

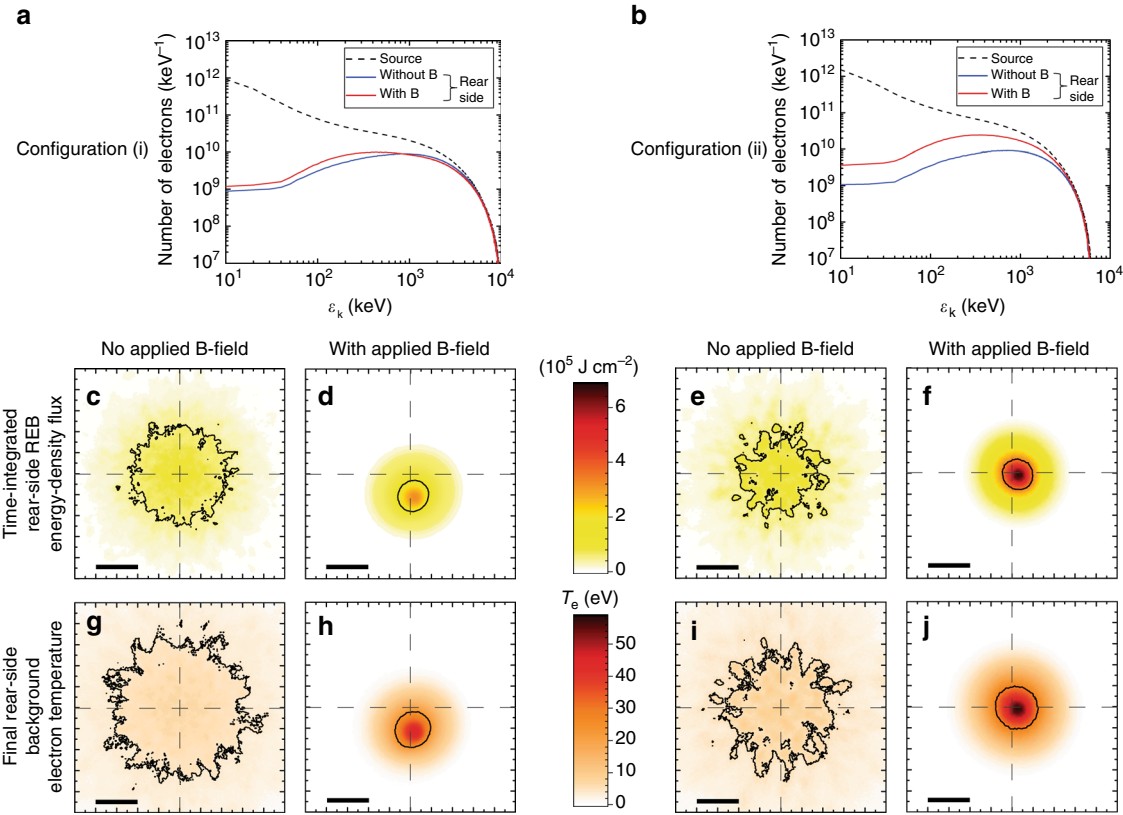

**Fig. 3** Relativistic electron beam features unfolded from transport simulations. The results, taken at the end of the 3D PIC-hybrid benchmarked simulation runs, are plotted for target-position configurations (i) on the left and (ii) on the right, without and with B-field. **a, b** Time-integrated relativistic electron beam (REB) energy spectra at the targets' front side (source, dashed lines) and rear side (full lines, red with imposed B-field, blue without B-field). **c–f** Time-integrated REB energy-density flux at the targets' rear surface. **g–j** Final background electron temperature at the targets' rear surface. In **c–j** the black bars stand for the 40 μm spatial scale, the contour lines correspond to the half-height of the signals and the crossed dashed lines indicate the position of REB injection at the targets' front surface

counterparts (symbols) for different values of the imposed B-field strength, in the case of target position configuration (ii). The changes in both pattern surface (Fig. 5a) and the CTR yield ratio with/without B-field (Fig. 5b) predicted by the simulations are consistent with both experimental metrics for B-field strengths at the coil center, $B_0$ (the plotted abscissa), ranging from 500 to 600 T. This is consistent with our experimental characterization of the B-field strength, $B_0 = 600 \pm 60$ T, in the specifically dedicated laser shots[42]. This set of simulation results highlights the improvement of the REB transport as a function of the B-field strength: For $B_0$ up to 200 T, the mitigation of electron filamentation is evidenced by a strong increase of the CTR yield (Fig. 5b), while not significantly changing the CTR pattern area (Fig. 5a). The yield ratio reaches a plateau for slightly stronger B-fields and increases again for $B_0 > 350$ T concurrently with a significant drop of the pattern surface. This is now a consequence of an efficient radial confinement of the majority of the REB electrons, expectable from the already mentioned guiding criterion: only at 350 T the Larmor radius of 2 MeV electrons (approximately the maximum energy of the REB spectra; see Fig. 3) finally drops to the REB initial radius, $r_0 \approx 25$ μm.

## Discussion

In conclusion, we succeeded to efficiently guide a laser-accelerated MeV electron beam through solid-density matter by imposing a 600 T B-field parallel to the electron beam propagation axis. The B-field was generated by using a coil target driven by a high-energy ns-laser interaction[42]. This B-field was driven 1 ns before the REB acceleration, providing a sufficient time for the

magnetization of the CH 60-μm-thick transport targets. According to benchmarked simulations accurately reproducing the experimental data in our best setup configuration, we found that the energy density transported by the fast electrons to the targets' rear surface and the peak background electron temperature increase, respectively, by factors of ≈5.3 and ≈5.9 compared to the case without imposed B-field. This enhancement in energy-density transport through dense matter is notable when compared to experiments based on the REB guiding by self-generated resistive B-fields[29–33]. Our experimental all-optical platform for strong B-field production and guided transport of laser-accelerated high-energy particles sets the ground for laboratory studies in regimes of matter opacities and equations of state at extreme temperatures. In the particular context of laser-fusion research, relevant experiments with target compression in magnetized conditions and magnetically guided REB should potentially optimize energy coupling to high-density cores of nuclear fuel[9,28,53–55].

## Methods

**Strong magnetostatic fields driven by laser.** Magnetic fields (B-fields) with peak value ≈600 T (±10%) and rise time of ≈1 ns were produced by the interaction of high-energy (500 J), 1-ns-duration laser pulses focused at $10^{17}$ W cm$^{-2}$ into Ni targets formed by two parallel disks, connected by a coil-shaped wire[42]. The target is charged by the laser interaction with the rear disk passing through a hole on the front disc: supra-thermal electrons of higher energy can escape from the potential barrier and are ejected from the interaction region. Eventually, a fraction of them is captured on the opposite, holed disk. Simultaneously, the coil-shaped wire reacts like an RL circuit and a discharge current produces a dipole-like B-field. The B-field at the center of the coil $B_0 \approx \mu_0 I/2a$ is related to the discharge current intensity $I$ looping in the target and to the coil radius $a$. $\mu_0$ is the vacuum permeability.

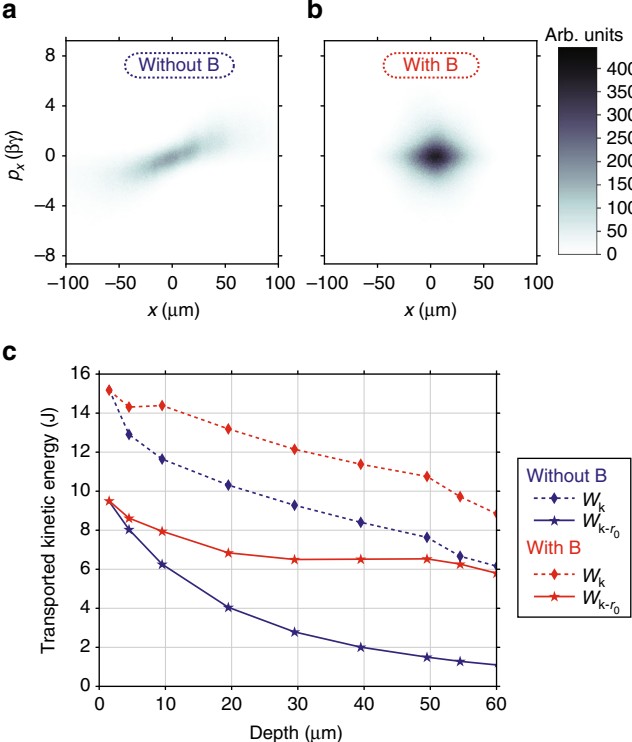

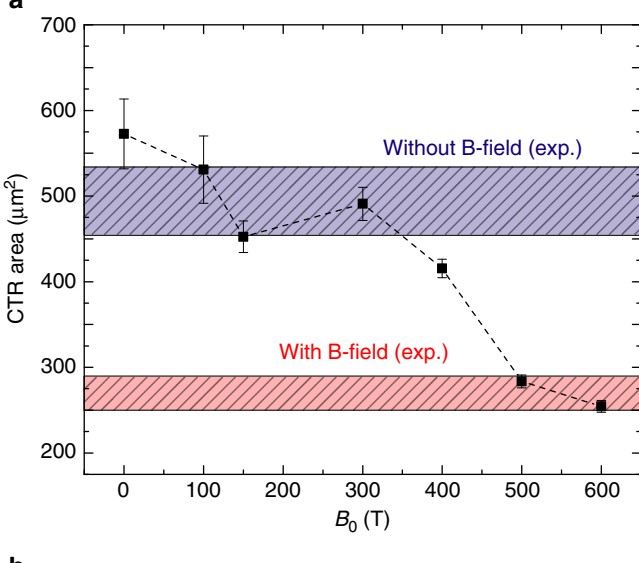

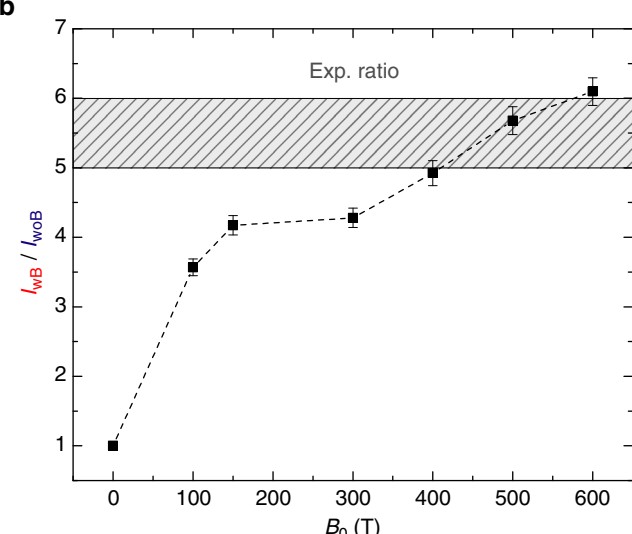

**Fig. 4** Relativistic electron beam phase space and transported energy. **a**, **b** Relativistic electron beam (REB) transverse phase space ($x$, $p_x$) at the target's rear surface, **a** without imposed B-field, **b** with imposed B-field. **c** Transported energy as a function of the propagation depth into the target, with (red) and without (blue) imposed B-field: total transported kinetic energy ($W_k$, diamonds) and its fraction within the initial REB radius $r_0$ centered on the target axis ($W_{k-r_0}$, stars). All the plots correspond to the transport-target in configuration (ii), positioned at the coil plane

The B-field strength rises monotonously during laser irradiation and then decays over a timescale of a few nanoseconds. The corresponding Super-Alfvénic currents can be theoretically explained according to the space charge neutralization and the magnetization of the plasma produced by the driver-laser between the disks[46]. In our experiment, using 250-μm-radius Ni coil targets, the spatial-integrated energy of the B-field at peak time corresponds to ≈4.5% of the driver laser energy and is distributed over a volume of ≈1 mm³ (see Supplementary Note 1 for further details).

**Simulations of relativistic electron beam transport**. PIC-hybrid simulations allow to describe REB transport in dense matter, where the injected beam current is modeled kinetically by a particle-in-cell (PIC) method and the neutralizing return current of background thermal electrons is described as an inertialess fluid[50,56,57]. Our simulation box corresponded to the transport-target dimensions, reproducing its CH-Cu structure in terms of background density and resistivity behavior as a function of the evolving background electron temperature due to REB-deposited energy. The background electron temperature is initiated at 0.1 and 1 eV, respectively, for the cases without and with B-field. The higher initial temperature in the later case accounts for the target pre-heating by intense X-rays issuing from the LP–laser interaction and the coil driven by the intense discharge current.

The REB source injected to the simulations at the target front surface, described by parameters given in the main text, is consistent with measurements from our previous experiments of REB-transport without exterior B-field, carried out in the same facility with equivalent SP-laser parameters and absolutely calibrated diagnostics[52].

Given the picosecond-timescale of the REB transport, very fast if compared to the nanosecond-scale evolution of the B-field strength or of its diffusion in the target, we assumed that the B-field distribution is constant over each simulation run-time. For the cases with applied external B-field, we only considered fully magnetized targets ($\Delta t = 1$ ns). The B-field spatial distribution inside the target is calculated as in vacuum with a 3D magnetostatic code[58], which is consistent with the experimental characterization of the B-field space-time evolution obtained with laser-driven coil targets[42].

The presented simulations assume no reflective conditions at the edges of the simulations box, which size corresponded to the real target size. We are specifically

**Fig. 5** Influence of the imposed B-field strength on the coherent transition radiation. **a** Synthetic coherent transition radiation (CTR) pattern surface size (at full-width-at-half-maximum) evolution with B-field strength at the coil center, $B_0$ (symbols). Red and blue bands identify the experimental CTR pattern sizes with and without applied B-field. **b** Ratio of synthetic CTR yield with/without applied B-field evolution with $B_0$ (symbols). The gray band identifies the experimental ratio. The images were analyzed as follows: Experimental images were first subtracted from background noise, which produces the uncertainty on the experimental yield ratio. Synthetic images are convoluted by $\sigma = 3 \pm 1$ μm s.d. Gaussian functions, matching the spatial resolution of the experimental diagnostic. Error bars on synthetic yield ratios ensue from the considered $\Delta\sigma = \pm 1$ μm. Both synthetic and experimental surface values are extracted by fitting CTR patterns by bi-dimensional Gaussian functions. Uncertainties on fit parameters arise from the pattern asymmetries

interested on unfolding the B-field effects over the first REB pass, that is on the forwardly directed energy transport. CTR measurements are adapted as the signals are the signature of mainly the first REB transit. Additional simulations with reflective conditions are presented in Supplementary Note 6 and Supplementary Fig. 5.

**Coherent transition radiation**. Coherent Transition Radiation (CTR) is produced by the REB crossing the target-vacuum boundary[19,47]. Its timescale, of the order of a few picoseconds, follows that of the fast electron flux envelope. Our experiment was mainly interested in studying the effects of the imposed B-field on the REB

energy forwardly transported into the target. CTR is an adapted diagnostic as it is mainly the signature of the first REB transit at the target's rear-side. The reasons are that CTR is mostly sensitive to the electrons of approximate MeV or higher energy, the higher-energy part of the REB spectrum[47,49], and that the emission relies on a sharp interface between the dense target and vacuum, that is no longer the case when a part of the beam electrons, mostly the lower-energy part, refluxes into the target due to electrostatic fields at the target edge's[59]: indeed the target surface expands into vacuum due to the REB energy deposition and induced heating[16].

The pulsed character of relativistic laser-acceleration mechanisms modulates longitudinally the REB current as a comb of periodic microbunches. The coherent interference of the transition radiation produced by the electron comb crossing the rear surface yields peak emissions at the spectral harmonics of the bunch frequency[48]. $2\omega_0$-light was selected by a 10 nm full-width-at-half-maximum (FWHM)-bandwidth interferometric filter centered at 532 nm. The CTR imaging system of the transport targets' rear surface was composed by two doublet lenses with an optical aperture of $f/9$, and was aligned on the equator plane looking at the target rear Cu surface with a 22.5° angle with respect to its normal. The optical system produced images with a magnification of ≈×20 with a spatial resolution of ≈7 μm FWHM. The streak camera was used with an open slit (≈5 mm) and the faster sweep speed of 0.5 ns/screen synchronized to the SP laser beam interaction, allowing to freeze the 2D pattern of the prompt CTR $2\omega_0$ emission from the target surface. As an extra precaution for reducing the noise level due to any spurious light from the coil target, only the central region of the imaged field, corresponding to the REB-transport target surface, was selected after the first lens.

The CTR data was used as benchmarking reference for the electron transport simulations coupled to a synthetic CTR-emission post-processor. As the hybrid transport code continuously injects particles during the laser pulse duration, we assumed that the periodic electron microbunches produced throughout the duration of the laser–plasma interaction are identical. The CTR is therefore calculated for one wavelength, 532 nm, and for a single electron bunch, yielding intensities in arbitrary units. More details are found in Supplementary Note 4.

**Data availability**. The authors declare that all data supporting the findings of this study on REB transport in magnetized solid-density targets are available within the paper and its Supplementary Information files. Other data concerning for example the characterization of the laser-driven magnetostatic fields were published elsewhere and are available from the corresponding author upon reasonable request.

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

## Acknowledgements

We gratefully acknowledge the support of the LULI pico 2000 staff during the experimental run. J.J.S. gratefully acknowledges fruitful discussions with L. Gremillet. This work was performed through funding from the French National Agency for Research (ANR) and the competitiveness cluster Alpha—Route des Lasers, project number TERRE ANR-2011-BS04-014. The authors also acknowledge support from the COST Action MP1208 'Developing the physics and the scientific community for Inertial Fusion' through three STSM visit grants. The research was carried out within the framework of the 'Investments for the future' program IdEx Bordeaux LAPHIA (ANR-10-IDEX-03-02) and of the EUROfusion Consortium and has received funding from the Euratom research and training programme 2014–2018 under grant agreement No. 633053. The views and opinions expressed herein do not necessarily reflect those of the European Commission. The Japanese collaborators were supported by the Japanese Ministry of Education, Science, Sports, and Culture through Grants-in-Aid for Young Scientists (Grant No. 24684044), Grants-in-Aid for Fellows by JSPS (Grant No. 14J06592), and the program for promoting the enhancement of research universities. Simulation work has been partially supported by the Spanish Ministry of Economy and Competitiveness (grant No. ENE2014-54960-R) and used HPC resources and technical assistance from BCS and CeSViMa centers of the Spanish Supercomputing Network.

## Author contributions

J.J.S. designed and executed the experiment as principal investigator, with the help from M.B.-G., C.B., P.F.-C., S.F., L.G., J.E.C., R.C., J.-L.D., M.E., S.K., J.-R.M., A.M., S.S., J.S. and Z.Z. and the engineering support from R.B., M.C., S.D., E.L. and F.S.; data analysis was performed by M.B-G., with contributions from P.F.-C., L.G., M.E., S.K., S.S. and Z.Z. and under the supervision of J.J.S., D.B. and S.F.; B-field diffusion modeling was performed by J.J.H., who also developed the PIC-hybrid code; the CTR post-processor was developed by C.B.; PIC-hybrid simulations were run and analyzed by M.B.-G. with supervision of J.J.S, C.B. and J.J.H.; targets were manufactured by G.S.; G.G., S.H., Ph.N., M.R., V.T.T. and N.W. contributed to the discussion of the results; M.B.-G. and J.J.S. led the manuscript writing; all the figures were prepared by M.B.-G.

## Additional information

**Competing interests:** The authors declare no competing financial interests.

