## [Peer Review File · Nature Communications]

Reviewers' comments:

Reviewer #1 (Remarks to the Author):

This was an interesting experiment and paper. The authors combined several recent techniques discussed in the last few years to develop an all optically driven system to confine hot electrons from a short pulse laser interaction with an external magnetic field generated by a laser driven coil. The authors report generating a magnetic field up to 600 T with this coil in the vicinity of the target which helps confine the hot electrons to a smaller volume. The authors present 3D PIC simulations showing the effect of the magnetic field on the electron beam and reasonable qualitative agreement with experiment. However, the authors only present data from a CTR imaging diagnostics which does not seem strong enough to support some of the claims in the paper, that energy density at and temperature of the rear surface have been increased by 5-6x that over the no-field case. The simulations suggest that is the case, but there is no supporting experimental evidence and thus I find that the conclusions (both text and abstract) in their current wording are too strong barring some additional experimental evidence. As the authors have demonstrated in the supplementary information, the correlation between CTR signal and electron beam density is not necessarily straightforward, and there are many effects which can modify the profile. Overall I thought the paper and the supplementary information was well-organized and written, physics overall was sound.

I have several questions regarding the simulations. The text makes mention that some of the particle losses are due to "being scattered out of the box", what is the simulation size and does it include a vacuum region, specifically whether refluxing can occur. The refluxing would have little effect on CTR signals but would play a heating role. What is the time in the simulation where the temperature profile presented in figure 3 is taken, and what fraction of the electron beam energy has been deposited. I have this question as well for figure 4, is it a single pass calculation or has the electron beam deposited or lost all of its energy? It's also not clear to me that the procedure for synthetic diagnostic is correct. My knowledge of CTR as a diagnostic is not very deep so correct me if I'm wrong. From the text, I interpret that you calculated the emission (which has incoherent and coherent components) for a single bunch with a charge q travel with some energy. You then assume that each macroparticle is a bunch, and attempt to coherently add these emissions including the time of flight variations but these individual macroparticles do not have the temporal coherency between bunches (which presumably are spaced at 2ω from $J \times B$) that real electron does so it's not clear to me that this is the same as the experimental measurement, can you elaborate further.

A few minor suggestions, on figures 1c and e, can you identify which surface is the "front" surface of the target. Additionally on figure 1e, it's difficult to see the black arrows on the blue background, I suggested recoloring them.

I think this paper has a lot of promise, presents interesting results from a challenging novel experiment, strongly supported by simulations. I agree with the authors that this platform has a lot of potential for both inertial confinement fusion and general HED science. I recommend the authors revise the manuscript and address some of these comments before a final decision is made.

Reviewer #2 (Remarks to the Author):

Understanding the effects that strong magnetic fields have on the properties of high energy density plasmas is currently an important research topic relevant to ICF. The manuscript deals with the transport of laser energy into a solid target by the laser produced electron beam. Efficiency of the energy transport is impaired by the large divergence of the beam. The authors claim that a longitudinal 600T magnetic field significantly improves the guiding of the electron beam. The energy density of the beam at the target rear side increases by a factor of 5. Guiding efficiency was estimated by coherent transition radiation emission on the rear target side at the second harmonic of laser radiation. This result if confirmed is new and interesting for the HEDP community. A 600T magnetic field was produced by a coil connected to two disc electrodes. One disc was irradiated by a 1ns pulse with energy of 500J. Fast electrons from the laser produced plasma reached the second disc and generated current and a magnetic field in the coil. A Supplementary Information file attached to the manuscript refers to the previous paper of the authors, Ref. [39] for details. For example, Figure 1(b) for measurement of the B-field was taken from [39].

Generation of a super-strong magnetic field is a key point in this research. The 800T magnetic field was presented in Ref. [39]. The field of 1500T was claimed in [38]. These publications were met with questions and critiques from the plasma physics community. The method of the laser driven magnetic field production was tested in the Laboratory for Laser Energetics at the University of Rochester but only a 40-50T magnetic field was measured at the coil center using proton radiography [see L. Gao et al., Phys. Plasmas 23, 043106 (2016)]. A simple model of laser driven generation of the magnetic field was published in a recent paper by G. Fixel et al., Appl. Phys. Lett. 109, 134103 (2016). It was shown that generation of 100T field is possible in this configuration but a 600-800T field is not realistic. For example, energy of a 800T magnetic field in one cubic mm is 250J compared to total laser energy of 500J. The magnetic field was measured in [39] at a distance of millimeters from the coil by the Faraday rotation diagnostic and centimeters by the magnetic probe. A magnetic field of 1-3 mT was really measured by the magnetic probe at the distance of 7cm [see Fig.1(b) in the Supplementary Information file] however it was approximated to 800T in the coil center. The 600-800T magnetic field claimed by the authors of the manuscript and Ref.[39] may be the result of an error. Articles of L. Gao and G. Fixel were not cited in the manuscript.

Since a magnetic field of 600T, in the presented configuration, is questionable, I do not recommend the manuscript for publication in the Nature Communications journal.

Reviewer #3 (Remarks to the Author):

Part 1:

The authors have demonstrated an unprecedented increase by a factor of roughly 5 in the energy density flux from laser-driven relativistic electron beams produced in and transported through a dense target. This was achieved by incorporating the use of a strong external, longitudinal B-field supplied by an optically-powered solenoidal coil target of their own making. The effect of the longitudinal B-field is to collimate and guide the accelerated electrons that are produced in the laser-target interaction through the entire bulk of the target. To accomplish this, the gyroradius of the electrons must be smaller than the spot size of the beam being produced, requiring a field of several hundred Tesla in strength. The optically-powered solenoidal magnetic device can generate quasi-static magnetic fields of up to 600 Tesla over nanosecond timescales, which is quite unique and not easily achieved through other means. This clever device is also compact (mm scale) and driven by a laser, making it potentially scalable, although it does require a significant amount of pulse energy of ~500 J,

roughly ten times more energy per pulse than in the laser used to produce the electrons.

The results presented in the article are unique and likely will lead to a significant step forward in several fields of research that take advantage of these laser-driven electron beams produced in solid targets, such as warm dense matter, inertial confinement fusion, and perhaps most obviously in proton and ion acceleration via methods like target normal sheath acceleration. Though the authors effectively state as much in their article, and though it certainly seems to be true, it is not made clear in any quantitative way how the current work will impact any one of these particular fields, and without that context, it may not be clear to the non-expert why the current work would truly be of interest. Certainly the electron beam itself is not unique or of particularly good quality in an abstract sense. Conventional accelerator guns can produce much higher quality electron beams, for example. Indeed, it is the fact that this beam is generated by a laser, originates from and is transported through a solid target, and that it has an enormous peak current of ~ 30 MA that makes it unique and of interest. If possible, it would be extremely interesting to at least see a numerical estimation (or even better, a simulation) of how the advances made in this work might improve the yield, angular distribution, etc. of a proton beam generated via transverse normal sheath acceleration, or some similar process. Such a connection would greatly improve the context of this significant advancement, emphasizing the importance of these results.

The quality and thoroughness of the work presented is excellent, the writing is superb, and the support of the conclusions that are drawn is quite convincing, overall. There is only one aspect of the work that begs a bit more careful attention and explanation, and that is the emittance analysis of the electron beam. It is claimed that the addition of the optically-powered solenoidal magnetic field reduces the emittance of the electron beam, but it is not clear why this would be the case. For a mono-energetic beam, the emittance could never be reduced by a solenoid field. This beam has a large energy spread ($\sim 100\%$), however, which means that its projected emittance would indeed increase even while propagating through vacuum, thus it is possible that such an effect could be reduced by the presence of a strong solenoid field. Space charge effects and collisional scattering in the target would also likely contribute to emittance growth and might be mitigated by the solenoid field. It should be clarified what effects contribute the most to the emittance growth of the beam without the presence of the solenoid field and how the solenoid field actually reduces these effects. As stated in the article, it leaves the impression that vacuum-like propagation on its own is the source of the beam's emittance growth without any reference to energy spread, space charge forces, or collisional scattering, and in that respect it is somewhat confusing. The solenoid field would of course prevent drift-like phase space skewing, but it would not reduce emittance (for a mono-energetic beam). For that matter, the emittance may not be really reduced much at all by adding the magnetic field; it depends on how the emittance is being calculated. If all macro-particles are being counted in the rms calculation, then tails can dominate the calculated emittance values. The tails might be significantly more pronounced without the solenoid field, giving the impression of significant overall emittance reduction when the field is present, when really it's just an insignificant number of tail particles that are the most affected. If that were the case, something like a 90% rms calculation might be a more appropriate calculation (and often is with realistic beams), where only the central 90% (or 95% or 99%) of the particles are counted. Although it would still not be entirely clear what the significance of the emittance would be if the entire energy range is included and indeed, perhaps it wasn't in the paper. That, too, should be clarified. A more meaningful emittance value might concern only the particles within, say, $\sim 10\%$ of the peak energy, for example. So the reviewer requests that the authors consider optimizing and clarifying their emittance calculations. In addition, the authors should provide a brief explanation of the physical processes that contribute most to the emittance growth of the beam without the presence of the solenoid field, and explain how the presence of the field reduces these effects.

Of final note, the article produces sufficient information for the reproduction of the analysis performed here, and indeed, it is likely that the techniques presented in this article will be adopted across multiple fields of research (proton/ion acceleration, warm dense matter, inertial confined fusion), given the substantial advancement in performance that has been demonstrated. The reviewer expects this work to be quite influential in nearly all research that utilizes laser-solid target interactions.

Part 2:

"I have reviewed Referee #2's comments and reconsidered the submitted manuscript in light of these comments. I studied the literature pointed to by Ref. #2 as well. In summary, I am also now convinced that the estimation of the on-axis B-field strength of ~600 Tesla in the manuscript does not appear to be sufficiently well substantiated, although there does appear to be sufficient evidence to believe that the on-axis B-field was at least ~100 Tesla.

Referee #2 makes the argument that the energy contained in the B-field would appear to be too large (~250 J), but the volume used to make this estimate was in fact rather large compared to the approximate volume of substantial magnetic field strength shown in the simulation depicted in Figure 3 of Reference 1 from the Supplemental Materials. That paper exactly describes the coil apparatus used in this experiment. By my own estimates, the relevant volume appears to be closer to $\pi \cdot (0.25 \text{ mm})^3$ instead of 1 mm^3 , and thus energy contained in the B-field ought to be closer to 10 J, which is only 2% of the energy from the initial 500 J long laser pulse, rather than 50%. That said, it still appears to be a very high efficiency compared to previous experiments that showed less than 1% efficiency, and it ought to be examined more carefully. So regarding this point, I think that there is reason to question the values put forth in the manuscript, though I do not believe they appear to be as egregious as Ref. #2 puts it.

Further, in the article by G. Fiksel referred to by Ref. #2 which is explicitly critical of previous work by the same group that wrote the submitted article, a major critique is that the high current required to generate the ~600 Tesla B-field--300 kA by their estimate, 240 kA by my estimate--would melt the narrow copper wire of the coil. The Onderdonk Equation, however, would tell us that the current would have to be sustained for ~10 us to melt this wire, whereas the high current lasts only for ~10 ns in the experiment. Thus I don't find this to be a valid argument against the plausibility of the ~600 Tesla B-field.

In conclusion, I do think that Ref. #2 raises valid concerns about the claim in the manuscript of having generated a ~600 Tesla B-field with the copper coil device. It seems plausible that the actual value may have been at least ~100 Tesla, but more evidence is needed to claim that it reached a higher value than that. It would be interesting at the very least to see if the CTR patterns in the data could be well replicated in simulation with smaller B-fields, going down to ~100 Tesla.

It should be noted, however, that the specific strength of the on-axis B-field was not itself the major claim of the paper. Rather, the claim of the paper was that the presence of the on-axis B-field provided significant focusing of the electron beam produced by the short pulse laser, thus increasing the beam energy density at the exit of the target by a factor of 5 as evidenced by the CTR intensity patterns, which is a very significant result. That there was at least a modestly strong B-field present, and that the beam energy density was increased by 5 when it was used is not in dispute. Thus I feel that if the authors can present a more credible claim for the high value of their magnetic field, or if they can show that the data appears consistent with a smaller and more easily supported field strength, then the paper is still worthy of publication in Nature Physics."

Response to the referees

NCOMMS-16-26237

M. Bailly-Grandvaux, J.J. Santos *et al.*

Guiding of relativistic electron beams in dense matter by laser-driven magnetostatic fields

We thank all the three referees for their revision and comments on our work, which significantly contributed to improve the submitted paper.

Below, the text in blue are the referees' reports, the text in black are our responses.

A list of changes in the article appears on the last page of this document.

Referee #1

"This was an interesting experiment and paper. The authors combined several recent techniques discussed in the last few years to develop an all optically driven system to confine hot electrons from a short pulse laser interaction with an external magnetic field generated by a laser driven coil. The authors report generating a magnetic field up to 600 T with this coil in the vicinity of the target which helps confine the hot electrons to a smaller volume. The authors present 3D PIC simulations showing the effect of the magnetic field on the electron beam and reasonable qualitative agreement with experiment. However, the authors only present data from a CTR imaging diagnostics which does not seem strong enough to support some of the claims in the paper, that energy density at and temperature of the rear surface have been increased by 5-6x that over the no-field case. The simulations suggest that is the case, but there is no supporting experimental evidence and thus I find that the conclusions (both text and abstract) in their current wording are too strong baring some additional experimental evidence. As the authors have demonstrated in the supplementary information, the correlation between CTR signal and electron beam density is not necessarily straightforward, and there are many effects which can modify the profile. Overall I thought the paper and the supplementary information was well-organized and written, physics overall was sound."

CTR was chosen as the main diagnostic for two main reasons: it is easy to freeze (a fast streaking with open slit works as fast-gated frame grabber of the early time emission, isolated from the later time thermal emission associated to the hydrodynamic target evolution) and its coherence insures (in particular at wavelengths corresponding to the laser harmonics) that it is mainly sensitive to the first transit of the REB through the radiating surface. This makes the data relevant for the effects of the imposed B-field over the first transit, in spite of the small size of the used transport targets.

We have also fielded a quartz spherical crystal looking at the Cu rear-side layer of the REB transport targets, in order to obtain 2D images of the Cu-K α fluorescence, attributable to the REB transverse pattern at the targets rear side. Unlike the CTR imaging technique, the K α diagnostic is not time-resolved and we could not freeze the K α emission from solely the first REB transit through the Cu layer. As the targets were very small transversely and longitudinally, of only 200 μ m-diameter and 60 μ m total thickness (category of the so-called *mass-limited targets*), we invariably obtained signals of diameter comparable to the target diameter, ensuing from multiple electron refluxing [see *e.g.* Quinn et al., Plasma Phys. Controll. Fusion **53**, 025007 (2011)]. There was no possible distinction between the results with and without B-field. In other words, the full Cu tracer became fluorescent on the time-integrated images, and the results were meaningless for our purpose which was to explore the B-field effects on the forwardly-directed energy transport by fast electrons.

While exclusively supported by the CTR imaging data (of excellent time and spatial resolutions), we consider outstanding the agreement between the data and the results of the PIC-hybrid transport simulations. We agree that CTR is a complex radiation mechanism that can only be fully understood by accurately reproducing the real experimental conditions and setup geometry. Our understanding of the CTR diagnostic response as a function of the experimental setup parameters is described in detail in the Supplementary Information and illustrated by Fig. 3 therein.

The confidence on the predictability level of our transport simulations coupled to the CTR postprocessor is even better funded upon the extra simulation results we decided to add now to the paper for variable B-field strength (Fig. 5 in the main text and Fig. 4 in the Supp. Material): the CTR pattern size-reduction and yield enhancement are clearly only compatible to an enough strong imposed B-field, above 500 T (in agreement with the guided criterion discussed in the paper) and the calculated CTR emission is highly sensitive to the transition between a diverging and a guided REB. *In fine*, the benchmarked transport simulations presented in the submitted paper allow to confidently extract information on the transported energy-density, its deposition and on the resulting bulk electron temperature.

This said, taking the referee comment into account, we have changed the wording of our claims in both the abstract and the main text.

"I have several questions regarding the simulations. The text makes mention that some of the particle losses are

due to “being scattered out of the box”, what is the simulation size and does it include a vacuum region, specifically whether refluxing can occur. The refluxing would have little effect on CTR signals but would play a heating role. What is the time in the simulation where the temperature profile presented in figure 3 is taken, and what fraction of the electron beam energy has been deposited. I have this question as well for figure 4, is it a single pass calculation or has the electron beam deposited or lost all of its energy?”

As now stated in page 4, the total simulation time was set to a maximum of 3.6ps, with $t=1.25$ ps corresponding to the peak REB flux at the front surface. The simulations assume no reflective conditions at the edges of the simulations box, which size corresponded to the real target size. It is worth reminding that, unlike in full-PIC simulations, the PIC-hybrid approach is only valid when the background electron density is significantly greater than the REB density, therefore does not allow to simulate vacuum regions around the target.

The spectra, time-integrated fast electron flux and temperature maps in Fig. 3 are taken at the end of the simulations. Also, the synthetic CTR images in Fig. 2 are calculated at the end of the simulation runs. Yet, we observed that they differ by less than 1% from the results at 2.3ps and 2.5ps, respectively for the magnetized and the unmagnetized runs (as now specified in the Supp. Information).

We are specifically interested on unfolding the B-field effects over the first REB pass, that is on the forwardly directed energy transport. CTR measurements are adapted as the signals are the signature of mainly the first REB transit. This is now clearly stated in the Methods section. All the simulation results, including those in Fig. 4, are a single pass calculation.

Nonetheless, we have now performed additional simulations with *artificial* reflective conditions at the target’s surfaces for equivalent magnetization states. These additional results, and the corresponding simulation assumptions, are now presented and commented in the Supplementary Information. The isochoric heating of small solid-density samples is indeed of interest for driving high-energy density matter states. As seen in Fig. 5 of Supp. Information, while the final temperature at the rear side rises due to electron recirculation (for both magnetized and unmagnetized conditions), the gain in bulk electron temperature remains of a factor 5 when imposing the B-field (as in the simulations without reflective conditions).

“It’s also not clear to me that the procedure for synthetic diagnostic is correct. My knowledge of CTR as a diagnostic is not very deep so correct me if I’m wrong. From the text, I interpret that you calculated the emission (which has incoherent and coherent components) for a single bunch with a charge q travel with some energy. You then assume that each macroparticle is a bunch, and attempt to coherently add these emissions including the time of flight variations but these individual macroparticles do not have the temporal coherency between bunches (which presumably are spaced at 2ω from JxB) that real electron does so it’s not clear to me that this is the same as the experimental measurement, can you elaborate further.”

The temporal coherency between the fast electron micro-bunches determines the characteristic peaked lines at the laser harmonics usually seen in the CTR spectra (see our previous works and data, e.g. Baton et al. PRL 2003, Popescu et al. Phys. Plasmas 2005, Santos et al. Phys. Plasmas 2007, Bellei et al. PPCF 2012). In the submitted work, our diagnostic acquisition is deliberately limited to a 10nm bandwidth around the laser second harmonic, by means of an interferometric filter (as described in the text). The acquisition bandwidth has been chosen as the most intense CTR emission line that we could observe in the visible spectral range. We didn’t measure the CTR spectrum. Yet, the calculated CTR yields (in arbitrary units) account for coherence effects associated to the full electron beam length and the momentum dispersion. In the presented units, the relative variation in synthetic CTR yield due to the imposed B-field is directly comparable to the equivalent variation on the experimental yields. The overall description of the CTR post-processor was improved in both the Methods and the Supp. Material.

“A few minor suggestions, on figures 1c and e, can you identify which surface is the “front” surface of the target. Additionally on figure 1e, it’s difficult to see the black arrows on the blue background, I suggested recoloring them.”

The front surface is identifiable in Figs. 1c) and e) by the pink spots. We added the following phrase to the figure caption to make it clearer: “The pink spots represent the REB source size at the targets irradiated surface.” We have changed the arrows in Fig. 1e) for a lighter colour.

“I think this paper has a lot of promise, presents interesting results from a challenging novel experiment, strongly supported by simulations. I agree with the authors that this platform has a lot of potential for both inertial confinement fusion and general HED science. I recommend the authors revise the manuscript and address some of these comments before a final decision is made.”

We thank Referee #1 for her/his interesting and positive comments and hope that our explanations and the additional information added to the paper have contributed to make it clearer.

“Understanding the affects that strong magnetic fields have on the properties of high energy density plasmas is currently an important research topic relevant to ICF. The manuscript deals with the transport of laser energy into a solid target by the laser produced electron beam. Efficiency of the energy transport is impaired by the large divergence of the beam. The authors clam that a longitudinal 600T magnetic field significantly improves the guiding of the electron beam. The energy density of the beam at the target rear side increases by a factor of 5. Guiding efficiency was estimated by coherent transition radiation emission on the rear target side at the second harmonic of laser radiation. This result if confirmed is new and interesting for the HEDP community. A 600T magnetic field was produced by a coil connected to two disc electrodes. One disc was irradiated by a 1ns pulse with energy of 500J. Fast electrons from the laser produced plasma reached the second disc and generated current and a magnetic field in the coil. A Supplementary Information file attached to the manuscript refers to the previous paper of the authors, Ref. [39] for details. For example, Figure 1(b) for measurement of the B-field was taken from [39].

Generation of a super-strong magnetic field is a key point in this research. The 800T magnetic field was presented in Ref. [39]. The field of 1500T was claimed in [38]. These publications were met with questions and critiques from the plasma physics community. The method of the laser driven magnetic field production was tested in the Laboratory for Laser Energetics at the University of Rochester but only a 40-50T magnetic field was measured at the coil center using proton radiography [see L. Gao et al., Phys. Plasmas 23, 043106 (2016)]. A simple model of laser driven generation of the magnetic field was published in a recent paper by G. Fixel et al., Appl. Phys. Lett. 109, 134103 (2016). It was shown that generation of 100T field is possible in this configuration but a 600-800T field is not realistic. For example, energy of a 800T magnetic field in one cubic mm is 250J compared to total laser energy of 500J. The magnetic field was measured in [39] at a distance of millimeters from the coil by the Faraday rotation diagnostic and centimeters by the magnetic probe. A magnetic field of 1-3 mT was really measured by the magnetic probe at the distance of 7cm [see Fig.1(b) in the Supplementary Information file] however it was approximated to 800T in the coil center. The 600-800T magnetic field claimed by the authors of the manuscript and Ref.[39] may be the result of an error. Articles of L. Gao and G. Fixel were not cited in the manuscript.

Since a magnetic field of 600T, in the presented configuration, is questionable, I do not recommend the manuscript for publication in the Nature Communications journal.”

We would like to point out the following:

- i) The peak B-field of 600 ± 60 T that we report appears to us as not questionable on the grounds claimed by Referee #2 (please see our detailed response to the 2nd report of Referee #3).
- ii) We don't claim the production of 1500 T B-field as in ref. [38] (now ref. [41]). Actually, only a few of us among the present collaboration consortium had participated to that preliminary experiment, whose merit was nevertheless to pave the ground for strong B-field generation with nowadays kJ-level laser capabilities and for more accurate characterizations in the following experiments (now refs. [42], [43] and [45]). In the conditions of the presently submitted article, the efficiency of laser energy conversion to magnetic-field energy with Ni coil targets was estimated to $4.5\% \pm 0.8\%$ (ref. [42]).
- iii) Our main point in this article is not the exact value of the imposed peak B-field, it is the effect of the B-field on REB electron transport.
- iv) Our results show a clear difference in the CTR patterns' size, shape and yield according to the magnetization conditions, and the benchmarked results from the REB transport simulations unfolded an unprecedented enhancement of a factor 5 on the transported energy-density.
- v) This enhancement is only compatible with an imposed axial B-field in the range of 500 to 600 T, as analysed from the comparison of the CTR experimental signals with simulation results for variable B-field strength. This analysis is particularly documented by the added Fig. 5 in the main text and Fig. 4 in the Supplementary Information.
- vi) References by Gao et al. and by Fiksel et al. mentioned by the referee are, a) for the former, an account of B-field measurements in laser irradiation conditions far from optimal and definitively different from those of our experiments (used frequency tripled laser drivers, not to mention their particular target geometry), b) for the later, an approach to the physics of the B-field generation in laser-driven coil targets that could explain Gao et al.'s results, but that, to our understanding, cannot accurately describe the laser driven current source between the disks in the high irradiance regime that we explored. The references mentioned by the referee correspond to a regime of magnetostatic field production much lower than ours, and are out of the scope of the submitted paper. Gao et al.'s paper is now cited in the introduction, along with the papers mentioned above, for their contribution to develop a platform for magnetostatic field generation from laser-matter interactions.

"The authors have demonstrated an unprecedented increase by a factor of roughly 5 in the energy density flux from laser-driven relativistic electron beams produced in and transported through a dense target. This was achieved by incorporating the use of a strong external, longitudinal B-field supplied by an optically-powered solenoidal coil target of their own making. The effect of the longitudinal B-field is to collimate and guide the accelerated electrons that are produced in the laser-target interaction through the entire bulk of the target. To accomplish this, the gyroradius of the electrons must be smaller than the spot size of the beam being produced, requiring a field of several hundred Tesla in strength. The optically-powered solenoidal magnetic device can generate quasi-static magnetic fields of up to 600 Tesla over nanosecond timescales, which is quite unique and not easily achieved through other means. This clever device is also compact (mm scale) and driven by a laser, making it potentially scalable, although it does require a significant amount of pulse energy of ~500 J, roughly ten times more energy per pulse than in the laser used to produce the electrons.

The results presented in the article are unique and likely will lead to a significant step forward in several fields of research that take advantage of these laser-driven electron beams produced in solid targets, such as warm dense matter, inertial confinement fusion, and perhaps most obviously in proton and ion acceleration via methods like target normal sheath acceleration. Though the authors effectively state as much in their article, and though it certainly seems to be true, it is not made clear in any quantitative way how the current work will impact any one of these particular fields, and without that context, it may not be clear to the non-expert why the current work would truly be of interest. Certainly the electron beam itself is not unique or of particularly good quality in an abstract sense. Conventional accelerator guns can produce much higher quality electron beams, for example. Indeed, it is the fact that this beam is generated by a laser, originates from and is transported through a solid target, and that it has an enormous peak current of ~30 MA that makes it unique and of interest. If possible, it would be extremely interesting to at least see a numerical estimation (or even better, a simulation) of how the advances made in this work might improve the yield, angular distribution, etc. of a proton beam generated via transverse normal sheath acceleration, or some similar process. Such a connection would greatly improve the context of this significant advancement, emphasizing the importance of these results."

Impact and applications of an enhanced relativistic electron beam propagating into solid matter are indeed numerous, as we tried to point out in the paper introduction. Yet, we consider that numerical estimations towards secondary processes go beyond the scope of the present paper. They will naturally find their place in forthcoming papers along with new data.

"The quality and thoroughness of the work presented is excellent, the writing is superb, and the support of the conclusions that are drawn is quite convincing, overall. There is only one aspect of the work that begs a bit more careful attention and explanation, and that is the emittance analysis of the electron beam. It is claimed that the addition of the optically-powered solenoidal magnetic field reduces the emittance of the electron beam, but it is not clear why this would be the case. For a mono-energetic beam, the emittance could never be reduced by a solenoid field. This beam has a large energy spread (~100%), however, which means that its projected emittance would indeed increase even while propagating through vacuum, thus it is possible that such an effect could be reduced by the presence of a strong solenoid field. Space charge effects and collisional scattering in the target would also likely contribute to emittance growth and might be mitigated by the solenoid field. It should be clarified what effects contribute the most to the emittance growth of the beam without the presence of the solenoid field and how the solenoid field actually reduces these effects. As stated in the article, it leaves the impression that vacuum-like propagation on its own is the source of the beam's emittance growth without any reference to energy spread, space charge forces, or collisional scattering, and in that respect it is somewhat confusing. The solenoid field would of course prevent drift-like phase space skewing, but it would not reduce emittance (for a mono-energetic beam). For that matter, the emittance may not be really reduced much at all by adding the magnetic field; it depends on how the emittance is being calculated. If all macro-particles are being counted in the rms calculation, then tails can dominate the calculated emittance values. The tails might be significantly more pronounced without the solenoid field, giving the impression of significant overall emittance reduction when the field is present, when really it's just an insignificant number of tail particles that are the most affected. If that were the case, something like a 90% rms calculation might be a more appropriate calculation (and often is with realistic beams), where only the central 90% (or 95% or 99%) of the particles are counted. Although it would still not be entirely clear what the significance of the emittance would be if the entire energy range is included and indeed, perhaps it wasn't in the paper. That, too, should be clarified. A more meaningful emittance value might concern only the particles within, say, ~10% of the peak energy, for example. So the reviewer requests that the authors consider optimizing and clarifying their emittance calculations. In addition, the authors should provide a brief explanation of the physical processes that contribute most to the emittance growth of the beam without the presence of the solenoid field, and explain how the presence of the field reduces these effects."

The referee is right in pointing out that the way we quantified the REB emittance is misleading. Following her/his suggestion, we selected three energy ranges REB within the REB energy spectrum, representative of a) the more numerous lower-energy component, b) of the REB population mean energy, and c) of the order of the T_{h} -parameter

describing the higher-energy component. The evolution of the emittance for each of these three components, as a function of the target depth, is now presented and discussed in the Supplementary Information.

(Given the length limit constraint, the space liberated in the main text is now used to present the REB-transport simulation results for variable B-field strength, compared to the experimental data, which corroborate the values of B-field characterized separately and previously published in our paper ref. [42].)

By analysing the propagation of the electrons of each of the considered energy ranges, we indeed concluded that the B-imposed field does not produce changes in the beam emittance except for the lower-energy bin, where the high diffusivity of the electrons in the solid-density propagation medium raises the particle divergence. The diffusion effect on the beam emittance is nevertheless mitigated for propagation in magnetized media.

"Of final note, the article produces sufficient information for the reproduction of the analysis performed here, and indeed, it is likely that the techniques presented in this article will be adopted across multiple fields of research (proton/ion acceleration, warm dense matter, inertial confined fusion), given the substantial advancement in performance that has been demonstrated. The reviewer expects this work to be quite influential in nearly all research that utilizes laser-solid target interactions."

Referee #3's second report

"I have reviewed Referee #2's comments and reconsidered the submitted manuscript in light of these comments. I studied the literature pointed to by Ref. #2 as well. In summary, I am also now convinced that the estimation of the on-axis B-field strength of ~600 Tesla in the manuscript does not appear to be sufficiently well substantiated, although there does appear to be sufficient evidence to believe that the on-axis B-field was at least ~100 Tesla.

Referee #2 makes the argument that the energy contained in the B-field would appear to be too large (~250 J), but the volume used to make this estimate was in fact rather large compared to the approximate volume of substantial magnetic field strength shown in the simulation depicted in Figure 3 of Reference 1 from the Supplemental Materials. That paper exactly describes the coil apparatus used in this experiment. By my own estimates, the relevant volume appears to be closer to $\pi(0.25 \text{ mm})^3$ instead of 1 mm^3 , and thus energy contained in the B-field ought to be closer to 10 J, which is only 2% of the energy from the initial 500 J long laser pulse, rather than 50%. That said, it still appears to be a very high efficiency compared to previous experiments that showed less than 1% efficiency, and it ought to be examined more carefully. So regarding this point, I think that there is reason to question the values put forth in the manuscript, though I do not believe they appear to be as egregious as Ref. #2 puts it.

Further, in the article by G. Fiksel referred to by Ref. #2 which is explicitly critical of previous work by the same group that wrote the submitted article, a major critique is that the high current required to generate the ~600 Tesla B-field--300 kA by their estimate, 240 kA by my estimate--would melt the narrow copper wire of the coil. The Onderdonk Equation, however, would tell us that the current would have to be sustained for ~10 μ s to melt this wire, whereas the high current lasts only for ~10 ns in the experiment. Thus I don't find this to be a valid argument against the plausibility of the ~600 Tesla B-field.

In conclusion, I do think that Ref. #2 raises valid concerns about the claim in the manuscript of having generated a ~600 Tesla B-field with the copper coil device. It seems plausible that the actual value may have been at least ~100 Tesla, but more evidence is needed to claim that it reached a higher value than that. It would be interesting at the very least to see if the CTR patterns in the data could be well replicated in simulation with smaller B-fields, going down to ~100 Tesla.

It should be noted, however, that the specific strength of the on-axis B-field was not itself the major claim of the paper. Rather, the claim of the paper was that the presence of the on-axis B-field provided significant focusing of the electron beam produced by the short pulse laser, thus increasing the beam energy density at the exit of the target by a factor of 5 as evidenced by the CTR intensity patterns, which is a very significant result. That there was at least a modestly strong B-field present, and that the beam energy density was increased by 5 when it was used is not in dispute. Thus I feel that if the authors can present a more credible claim for the high value of their magnetic field, or if they can show that the data appears consistent with a smaller and more easily supported field strength, then the paper is still worthy of publication in *Nature Physics*."

Here we will discuss the evidence of 500 to 600T B-field strength directly from the results of relativistic electron transport.

Yet before and for sake of clarity (as some confusion appears in the referees' reports), allow us to summarize our previous results on the characterization of laser-driven quasi-static B-fields, published in two papers: Santos *et al.*, *New J. Phys* **17**, 083051 (2015) (now ref. [42]) and Law *et al.*, *App. Phys. Lett.* **108**, 091104 (2016) (now ref. [43]).

The first of these two papers [Santos *et al.*, 2015] reported results obtained in two runs at the LULI2000 facility, at Ecole Polytechnique, France. From the first run, we reported maximum B-field amplitude at the coil centre of 800

T, 600 T and 150 T for 250 μ m-radius coils of Cu, Ni or Al respectively, with 6% accuracy. These values correspond to peak coil-discharge currents of 340 kA, 255 kA and 64 kA respectively, according to 3D magnetostatic simulations. The current and B-field values at the coil centre were obtained from 3D magnetostatic extrapolations from the measurements of the B-field time derivative at a few cm from the target-coil, using high-frequency bandwidth B-dot probes. The time of the peak values was consistent with the 1ns-duration of the driver laser. The volume integration of the magnetic energy density for such B-field strengths and spatial distributions yields $8.3\% \pm 1.5\%$, $4.5\% \pm 0.8\%$ and $0.35\% \pm 0.05\%$ of the 500J laser energy, for the Cu, Ni and Al targets respectively.

On a second run at the LULI2000 facility we scrutinized the use of Ni targets, and have demonstrated the repetitiveness of the B-field peak value (600T) and rising time (1ns). Proton-deflectometry data, measuring directly the fields at the coil region, confirmed the strength of the B-field at the coil centre up to ≈ 100 T at 0.35 ns, that is still before the time of the peak B-field. The subsequent time-evolution of the proton-deflectograms – showing decreasing and increasing deflections respectively for probing protons and relativistic electrons emitted from a backlighter foil target – gave evidence of progressive magnetization of ns-laser driven plasma electrons near the coil region. Such localized accumulation of negative charge could explain the proton-focusing electrostatic effect (along with defocusing of the probing relativistic electrons), added to the magnetic deflections.

In a subsequent experiment, carried out at the Gekko XII facility at LLE, Univ. Osaka, Japan [Law *et al.*, 2016], the setup for proton-deflectometry measurements was improved by mechanically shielding the coils from the laser interaction region between the disks and from the plasma driven by such interaction. In consequence, peak B-fields of ≈ 600 T at the coil centre of in Ni targets were extrapolated from B-dot probe measurements at several cm, consistently with the values directly measured at the coil by proton-deflectometry (assuming, of course, only B-field effects on the observed proton deflections). These more recent results consolidated the capacity to drive and characterize B-fields of several hundred Tesla, as well as our protocol to extrapolate B-field at the coil centre from distant B-dot probe measurements. By the above we reaffirm that our characterization of the B-field time evolution for Ni targets, with 600 T peak value at ≈ 1 ns after start of the driver laser irradiation (as used in the submitted electron transport experiment) is robust within the reported 10% uncertainty on the field strength and of ± 100 ps on the time position [Santos *et al.*, 2015].

Beyond this, we have also recently developed a quasi-analytical model describing the physical mechanism of the ns-laser interaction, the consequent charging of the irradiated-disk, the establishment of a current source between the disks and of the discharge current through the connecting coil-shaped wire (now ref. [46]). Our model explains why it is physically possible to produce coil-discharge currents exceeding 100 kA. The maximum delivered current is proportional to the supra-thermal electrons *temperature*, to which the main control parameter is the driver laser irradiance $I\lambda^2$. The use of higher intensities (tight focusing) and longer wavelengths (as in our experiments, compared to other data in the literature) improve the target performance providing more energetic electrons capable to maintain high currents at the large potential jumps developing between the target's disks.

Yet, as the referee pointed out, the specific strength of the on-axis B-field is not the major claim of the presently submitted paper. Our main point here is that the strong B-field imposed externally to the REB transport in solid-density targets provided significant enhancement on the electron beam energy-density flux at the depth of several tens of microns!

The reported experimental results on relativistic electron beam (REB) transport were obtained during the same experiments carried out at LULI described above and reported in ref. [42]. We used exclusively Ni coil-targets for the REB-transport shots to avoid mixing x-ray signatures from the coil and from the transport-targets' Cu rear-layer. The B-field measurements presented in Fig. 1 of the Supplementary Information correspond indeed to the same data presented in ref. [42]. The REB-transport shots were performed alternating with shots devoted to characterize the B-field: for example, the same ps laser beam was used for REB generation in the former, and for driving the probing proton beam in the latter. The type of shot could be changed straightforward in a shot-to-shot manner by simply shifting back and forth by 5 mm the longitudinal position of the focusing off-axis parabola. The quality of the laser spot was controlled and optimized at the corresponding focusing plane before each shot.

Joining the referee suggestion, we had indeed performed REB-transport simulations for different strengths of the B-field assuming a similar spatial distribution, that is, we divided by a constant factor the B-field maps in every position. The results have now been added to the article: Fig. 5 in the main text and Fig. 4 in the Supplementary Information. The later of these figures shows the evolution of the synthetic CTR pattern and of the corresponding REB-flux at the target's rear surface for variable values of the B-field strength at the coil centre, B_0 . Graphs in the former of the added figures summarize the results from these extra simulations for the two quantities that we used to compare simulations with the experimental data: a) the equivalent surface of the CTR pattern at FWHM and b) the ratio of the CTR peak signals with B-field over the case without B-field. As expected from the guiding criterion where the magnetic guiding becomes effective only if the REB mean Larmor radius (for the mean kinetic energy of the REB source) becomes comparable to the initial radius of the injected REB, the simulations (symbols) can only simultaneously reproduce both experimental quantities (horizontal shaded bars) for B_0 between 500 and 600 T. For $B_0 \leq 300$ T, the CTR pattern surface is comparable to the experimental case without imposed B-field.

It is worth to note that the $100\mu\text{m}$ thick Ta-shield used to handle the transport targets [see setup in Fig. 1a) of the main text] has enabled protection from hard X-rays and particle emission from the driver ns-laser interaction region.

In conclusion, the REB transport results and in particular the contrast in the CTR images between the situations with and without B-field, along with the unfolded factor ≈ 5 enhancement on the energy-density flux at the targets' rear side, are a supplementary, albeit indirect, evidence of the B-field strength in agreement to the more direct characterizations described above and published in our paper ref. [42].

We thank Referee #3 for her/his two reports and her/his interest and positive attitude regarding our work and the presented results.

LIST OF CHANGES:

- Replaced “magnetic fields” by “magnetostatic fields” in the title, which more accurately describes our experimental configuration.
- Our claims on the scientific results have been rephrased in the abstract and the main text: the enhancement of the energy-density flux in the magnetized transport case is unfolded from simulations benchmarked from the experimental data.
- A few relatively recent references have been added to the introduction:
 - o Arefiev et al 2016 on the numerical work about the effects of imposed longitudinal B-field on the TNSA acceleration of protons.
 - o Vaisseau et al 2017 on the shear current resistive effects on the REB self-guiding
 - o Gao et al 2016 and Goyon et al 2017 on recent characterization by fast-proton deflectometry of laser-produced magnetostatic fields in loop targets.
- Legend of Fig. 1 b) and d) more clearly states that the size of the pink spots corresponds to the considered size of the REB source in the hybrid-PIC electron transport simulations.
- Arrows in Fig. 1 e) are now in white to improve its visibility.
- We added the phrase “CTR is a non-linear emission mechanism providing the signature of relativistic electrons at their first transit through the target's rear surface.” (and references) to the part introducing the CTR results. We want by this emphasize that our experiment was focused on measuring the effects of the imposed B-field over the first transit of the laser-accelerated relativistic electron beam.
- Fig. 3 c)-f) plot now the time-integrated REB flux at the targets' rear surface, instead of the REB density. Figure caption and main text have been modified accordingly.
- The 3.6ps duration of the all the transport simulation runs is indicated.
- The numerical study of the dependence of the REB behaviour on the imposed B-field strength has been added to the article. In the main text, the CTR pattern size and yield ratio *with / without B-field* are compared to the experimental measurements, emphasizing that the observed effects can only be replicated in the simulations when using a B-field in the range of 500 to 600 T (Fig. 5). Details on the numerical results are presented in the Supplementary Information and Fig. 4 therein.
- For sake of the needed space, the results and discussion on the evolution of the REB emittance have been removed from the main text (and from Fig. 4) and are now presented in the Supplementary Information. The discussion has been improved following the advice of Referee #3.
- We have also included in the Supplementary Information results and a short discussion on extra REB transport simulations with reflective conditions at the target's edges.

REVIEWERS' COMMENTS:

Reviewer #1 (Remarks to the Author):

Between the additional supplementary material, the changing of wording on the heating claim, and the inclusion of figure 5 to the document, I can recommend publication with only minor changes. While I had some concerns on the strength of the magnetic field (as brought up by the other reviewers), I can find no particular mechanism or law that rules out the generation of field of this magnitude nor is the exact value, the focus of this paper. Recent results from Goyon et al. PRE 95, 033208 showed 1/3 field with a coil twice as large, so these fields are certainly plausible. The authors have shown experimentally guiding of the electron beam over 60 μm , and reproduced trends with PIC simulations for a variety of shot conditions. Paper is well written, and I believe will be very interesting to the HED community.

Minor change Figure 1 c/e, pink spot is still difficult to discern from the B-field colorscale, I recommend a change to something with more contrast.

Reviewer #3 (Remarks to the Author):

The authors did a fine job in addressing the issues and questions that were raised by the referees. I have no further reservations in recommending it for publication.

Response to the referees

NCOMMS-16-26237

M. Bailly-Grandvaux, J.J. Santos *et al.*

Guiding of relativistic electron beams in dense matter by laser-driven magnetostatic fields

We thank the referees for their final assessment on our work.

Below, the text in blue are the referees' reports, the text in black are our responses.

Referee #1

Between the additional supplementary material, the changing of wording on the heating claim, and the inclusion of figure 5 to the document, I can recommend publication with only minor changes. While I had some concerns on the strength of the magnetic field (as brought up by the other reviewers), I can find no particular mechanism or law that rules out the generation of field of this magnitude nor is the exact value, the focus of this paper. Recent results from Goyon et al. PRE 95, 033208 showed 1/3 field with a coil twice as large, so these fields are certainly plausible. The authors have shown experimentally guiding of the electron beam over 60 μm , and reproduced trends with PIC simulations for a variety of shot conditions. Paper is well written, and I believe will be very interesting to the HED community.

Minor change Figure 1 c/e, pink spot is still difficult to discern from the B-field colorscale, I recommend a change to something with more contrast.

We changed the color of the spots representing the size of the relativistic electron beam (REB) source to light-blue. We printed out Fig. 1 in both color and black/white and verified, for both prints, the improvement in discernibility in respect to the colorscale of the B-field maps.

Referee #3

The authors did a fine job in addressing the issues and questions that were raised by the referees. I have no further reservations in recommending it for publication.